# Development and Characterization of Yellow Passion Fruit Peel Flour (*Passiflora edulis f.* flavicarpa)

**DOI:** 10.3390/metabo13060684

**Published:** 2023-05-25

**Authors:** Maria Clara Coutinho Macedo, Vinícius Tadeu da Veiga Correia, Viviane Dias Medeiros Silva, Débora Tamires Vitor Pereira, Rodinei Augusti, Júlio Onésio Ferreira Melo, Christiano Vieira Pires, Ana Cardoso Clemente Filha Ferreira de Paula, Camila Argenta Fante

**Affiliations:** 1Departamento de Alimentos, Faculdade de Farmácia, Campus Belo Horizonte, Universidade Federal de Minas Gerais, Belo Horizonte 31270-901, Brazil; 2Departamento de Ciências Exatas e Biológicas, Campus Sete Lagoas, Universidade Federal de São João del-Rei, Sete Lagoas 35702-031, Brazil; 3Departamento de Engenharia e Tecnologia de Alimentos, Faculdade de Engenharia de Alimentos, Universidade Estadual de Campinas, Campinas 130862-862, Brazil; 4Departamento de Química, Universidade Federal de Minas Gerais, Belo Horizonte 31270-901, Brazil; 5Departamento de Engenharia de Alimentos, Campus Sete Lagoas, Universidade Federal de São João del-Rei, Sete Lagoas 35702-031, Brazil; 6Departamento de Ciências Agrárias, Campus Bambuí, Instituto Federal de Educação, Ciência e Tecnologia de Minas Gerais, Bambuí 38900-000, Brazil

**Keywords:** functional food, bioactive compounds, *Passiflora edulis*, agricultural waste, sustainability

## Abstract

In this study, the peels of the yellow passion fruit (*Passiflora edulis f.* flavicarpa) were used to develop a flour that was evaluated in terms of its physicochemical, microscopic, colorimetric, and granulometric characteristics, its total phenolic compound and carotenoid contents, and its antioxidant capacity. Fourier Transform Infrared (FTIR) spectroscopy measurements were employed to investigate the constituent functional groups, compounds’ chemical profiles were assessed by Paper Spray Mass Spectrometry (PS-MS), and the compound’s chemical profiles were evaluated by Ultra-Performance Liquid Chromatography (UPLC). This flour presented a light color, heterogeneous granulometry, high carbohydrate, carotenoid, and total phenolic compound contents with high antioxidant capacity. Scanning Electron Microscopy (SEM) showed a particulate flour, which is supposed to contribute to its compactness. FTIR demonstrated the presence of functional groups corresponding to cellulose, hemicellulose, and lignin, constituents of insoluble dietary fiber. The PS-MS analysis suggested the presence of 22 substances, covering diverse component classes such as organic, fatty, and phenolic acids, flavonoids, sugars, quinones, phenylpropanoid glycerides terpenes, and amino acids. This research demonstrated the potential of using Passion Fruit Peel Flour (PFPF) as an ingredient for food products. The advantages of using PFPF comprise the reduction of agro-industrial waste, contribution to the development of a sustainable food system, and increment of food products’ functional profile. Moreover, its high content of several bioactive compounds can benefit consumers’ health.

## 1. Introduction

Several agro-industrial industries generate tons of waste, either solid (e.g., seeds, peels, pulps) or liquid. However, it is important to note that this biodegradable waste can be used as a source of natural components of interest [1]. For instance, the yellow passion fruit peel (*Passiflora edulis* f. flavicarpa) is an example of waste from the food industry, which presents high contents of phenolic compounds and meaningful antioxidant capacity [2]. Several studies have shown the health benefits of using passion fruit peels for human consumption, such as cholesterol reduction, weight gain control, improvement of intestinal transit in constipated individuals, and glycemic control for diabetic patients [3,4,5].

The passion fruit is representative of the Passifloraceae family, where the *Passiflora* genus includes more than 500 plant species. The *P. edulis*, one of the most consumed species worldwide, is characterized as a tropical plant well-adapted to hot environments [6]. According to Santos et al. [7], Brazil is the world’s largest producer of passion fruit. The Brazilian Institute of Geography and Statistics (IBGE) [8] reported that Brazil’s passion fruit production reached more than 680 thousand tons in 2021. It is important to mention that its peel represents more than 60% of the passion fruit mass [9]. Therefore, considering the passion fruit 2019 annual production, when approximately 95% of the production was targeted to juice and pulp industries [10], almost 390 thousand tons of passion fruit peel were generated in the fruit processing industries.

Although Passion Fruit Peel Flour (PFPF) has been used in the preparation and nutritional enrichment of various food products such as corn flour, fermented milk, biscuits, extrudates, dietary cookies, and cakes [3,11,12,13], most of the studies on the characterization of PFPF are limited to present only primary results using spectrophotometric methods. Coelho et al. [14] evaluated the technological properties of PFPF, such as its capacity to be used as a thickener, stabilizer, emulsifier, and gelling agent. Rosario et al. [15] also assessed the technological and functional potential of PFPF but not covering a substantial identification and quantification of the product’s bioactive compounds. On the other hand, Cazarin et al. [16] evaluated PFPF centesimal composition and antioxidant capacity. More detailed information regarding PFPF phenolic compounds profile and chemical composition is needed [17].

Therefore, this study aimed to characterize thoroughly PFPF samples, growing a databank that may be used for researchers looking for PFPF applications as food ingredients or developing new food products. In this study, we obtained the PFPF chemical profile by Ultra-Performance Liquid Chromatography (UPLC) and identified the major phenolic compounds through the ultrafast ionization technique Paper Spray Mass Spectrometry (PS-MS). Furthermore, physico-chemical analyses, colorimetric and granulometric determinations, microscopy, and evaluations of total phenolic compounds and carotenoids contents were conducted. Finally, diverse antioxidant capacity determination methods were employed to contribute to the characterization of the flour obtained from the passion fruit peels.

## 2. Materials and Methods

### 2.1. Chemicals and Reagents

Folin–Ciocalteau reagent and Iron (III) chloride hexahydrate (FeCl_3_·6H_2_O) were acquired from Dinâmica (São Paulo, Brazil). Methanol (Merck, Darmstadt, Germany) and formic acid (Panreac, Barcelona, Spain) were HPLC grade. 2,4,6-Tris(2-piridil)-s-triazina (TPTZ), 2,2-difenil-1-picril-hidrazil (DPPH), 6-Hidroxi-2,5,7, 8-tetrametilcromano-2-carboxylic acid (Trolox), 2,2′-Azino-bis(3-ethylbenzothiazoline-6-sulfonic acid) diammonium salt (ABTS), potassium persulfate, gallic acid, catechin hydrate, chlorogenic acid, caffeic acid, ellagic acid hydrate, and e quercetin were supplied by Sigma-Aldrich (St Louis, MO, USA). Potassium carbonate and Deionized water were purchased from Synth (Diadema, Brazil) and obtained from a Milli-Q (Millipore, Billerica, MA, USA) purification unit, respectively.

### 2.2. Material

Yellow passion fruit peels (*Passiflora edulis f.* flavicarpa) were acquired at a popular market in Sete Lagoas, State of Minas Gerais, Brazil.

### 2.3. Obtaining Passion Fruit Peel Flour

The *P. edulis* peels were transported to the laboratory (Food Engineering Department, Federal University of São João del-Rei, Sete Lagoas, Brazil) right after the acquisition. Then, the material was cleaned in flowing water, cut, evenly distributed on aluminum trays, and dehydrated in an oven with forced air circulation (400-TD, Ethik Technology, Vargem Grande Paulista, Brazil) at 60 ± 5 °C for 24 h. Afterward, the peels were fragmented in a blender (Philips Walita, São Paulo, Brazil) and sieved (32 mesh, 500 mm, Bertel Indústria Metalúrgica Ltd., Brazil). The PFPF was placed in glass flasks and stored at −18 °C for further analysis.

### 2.4. Color

The flour color was evaluated using a spectrophotometer (Konica Minolta, CM-2300d, Tokyo, Japan) according to the CIELab scale. The color parameters were *L** (brightness), *a** (red to green), and *b** (yellow to blue). *A** and *b** values were also used to calculate the parameters *h** (hue) and C* (chroma or color intensity) [18]. All evaluations were performed in triplicate.

### 2.5. Proximate Composition and Starch Content

The PFPF proximate composition analyses were performed according to the Association of Official Analytical Chemists’ (AOAC) [19] methods with adaptations. Moisture content was determined at 105 °C in an oven with forced air circulation (320-SE, Fanem, Guarulhos, Brazil) until constant weight. The lipid content was determined using a Soxhlet extractor (SL-202, Solab, Piracicaba, Brazil). The total protein content (N × 6.25) was determined by the Kjeldahl method. The ash content analysis was carried out by incineration in a muffle (2629, Fornitec, São Paulo, Brazil) at 550 °C. The carbohydrate content was determined by difference. The results are expressed as g/100 g sample on a dry matter basis (DMB). The starch content was determined according to AOAC methods [19]. All the evaluations were performed in triplicate.

### 2.6. Fourier Transform Infrared (FTIR) Spectroscopy

FTIR spectroscopy analysis was performed on a spectrometer (IRAffinity-1, Shimadzu, Kyoto, Japan), with a deuterated triglycine sulfate doped with L-alanine (DLaTGS) detector and attenuated total reflectance (ATR) accessory with a crystal of zinc. The PFPF spectrum was obtained in the range of 4000 to 500 cm^−1^ (20 scans per experiment at the resolution of 4 cm^−1^).

### 2.7. Scanning Electron Microscopy (SEM)

The microstructure of the PFPF surface was analyzed using a Scanning Electron Microscope with Energy Dispersive X-ray Detector (models Leo 440i and EDS 6070, LEO Electron Microscopy/Oxford, Cambridge, UK). The samples were dried and kept under vacuum for 24 h before analysis. Then, they were coated with a layer of gold with an estimated thickness of 200 A° in a Sputter Coater spray applicator (K450, Emitech, Kent, UK). Analyses of sample surfaces were performed under vacuum, using an accelerating voltage of 10 kV and magnifications of 500×, 800×, and 2000×.

### 2.8. Granulometry

The granulometry determination and particle size distribution were carried out according to the method described by Martino et al. [20], where 100 g of flour was sieved in a sieve shaker (Lucadema, São José do Rio Preto, Brazil) for 10 min, using speed controller level 10, in a total of 11 Brazilian Association of Technical Standards (ABNT) standard vibrating sieves. The sieves’ mesh size ranged from 125 to 4000 µm, and the flour retained in each sieve was weighed and expressed as a mass percentage (%).

### 2.9. Preparation of PFPF Extracts

The PFPF extract preparation was carried out as described by Rufino et al. [21]. Thus, 2.5 g of PFPF was mixed with 10 mL of methanol/water solution (50:50, *v*/*v*) and left to rest for 1 h. After this period, the samples were centrifuged at 3493× *g* for 22 min (206BL Excelsa II, Fanem, Guarulhos, Brazil) when the supernatant was recovered. Subsequently, 10 mL of acetone/water solution (70:30, *v*/*v*) was added to the remaining residue and the process was repeated under the same conditions previously described. Both supernatants were mixed in a 25 mL flask, filling the empty volume with distilled water.

### 2.10. Total Phenolic Compounds and Antioxidant Capacity

The samples’ total phenolic compound (TPC) content was determined using the Folin–Ciocalteu method [22]. Therefore, 0.5 mL of the obtained extract was diluted in 0.5 mL of distilled water. An aliquot of 100 µL of the diluted extract was mixed with 650 µL of methanol solution (50%), 3.5 mL of distilled water, and 250 µL of Folin–Ciocalteu reagent. The mixture was left to rest at room temperature for 3 min. Afterward, 500 µL of sodium carbonate solution 7.5% (*m*/*v*) was added to the mixture that was homogenized in a tube agitator. Subsequently, the samples and the standards were kept for 60 min at room temperature away from light. The absorbance was measured at 750 nm on the spectrophotometer. A methanol solution of 50% (*v*/*v*) was used to prepare the standard gallic acid solution. The results are expressed as gallic acid equivalents (mg GAE·g^−1^ of sample on a DMB).

To evaluate the antioxidant capacity according to the DPPH method [19], three varying volumes of extract (50, 100, and 250 µL) were diluted with distilled water to the volume of 1 mL. Diluted extract aliquots of 100 µL were mixed with 5 mL of DPPH (40 mg.L^−1^) methanolic solution and incubated at 35 °C for 4 h. Subsequently, the readings were performed at 517 nm, using distilled water as blank. The Trolox standard solution (0.5 mg.mL^−1^) was used to build the analytical curve (Trolox mass x absorbance) at 0.01 to 0.04 mg. The results are expressed as Trolox equivalents (µM TE·g^−1^ of sample on a DMB).

To determine the antioxidant capacity by ABTS*+ capture [21], three varying volumes of extract (100, 250, and 500 µL) were diluted with distilled water to the volume of 1 mL. The ABTS*+ radical was prepared from the ABTS solution mixture (7 mM) with potassium persulfate solution (140 mM). After 16 h resting at dark, the solution of ABTS radical was diluted with ethanol (95%) until the absorbance of 0.700 ± 0.020 nm measured at 734 nm. Then, 3 mL of this solution was mixed with 30 µL of each diluted extract. After tube agitator homogenization, the absorbance was read after a 6 min resting period. The Trolox standard solution at 0.5 mg mL^−1^ was used to build the analytical curve in the range of 100 to 200 µM. The results of antioxidant capacity are expressed as Trolox equivalents (µM TE g^−1^ of sample on a DMB).

To use the FRAP method [21], the reagent solution was prepared by mixing buffer acetate solution pH 3.6 (0.3 M), TPTZ solution (10 mM), and ferric chloride solution (20 mM) at 10:1:1, respectively. Three varying volumes of extract (50, 75, and 100 µL) were diluted with distilled water to the volume of 1 mL. Then, 90 µL of each prepared diluted extract, 270 µL of distilled water, and 2.7 mL of FRAP reagent were added to assay tubes covered with foil sheets in triplicate. The solutions were homogenized and heated to 37 °C for 30 min. Subsequently, the absorbance reading was conducted at 595 nm. The FRAP reagent was used as a blank. The ferric sulfate solution (2 mM) was used as the standard to build the analytical curve (from 250 to 2000 µM). The results are expressed as µM of ferric sulfate g^−1^ of sample on a DMB.

### 2.11. Carotenoids

The quantification of total carotenoids in the extracts was conducted using spectrophotometry at 450 nm, according to the method described by Rodriguez-Amaya [23]. The results are expressed as μg of carotenoids/g on a DMB, and the concentration was calculated using Equation (1):CT (μg carotenoids/g sample) = (A × V × 104/E1%1 cm × m)(1)
where A = absorbance at 450 nm, V = final sample volume (mL), m = sample mass (g), E1%1 cm = the value of 2592 corresponding to the extinction coefficient of β-carotene in petroleum ether.

### 2.12. Determination of the Profile of Phenolic Compounds

The PFPF’s major phenolic compounds were determined following the chromatographic method described by Eça et al. [24]. The extracts were filtered using a nylon syringe filter (0.22 μm) and injected into UPLC (Waters, Acquity UPLC^®^ Class, Milford, MA, USA) equipped with a UV diode-array detector, quaternary pump, degassing system, and an automatic sampler. The collected data were processed using the Empower software. The UPLC was equipped with the Acquity UPLC^®^ BEH C18 column (2.1 × 100 mm; 1.7 µm, Waters, Milford, MA, USA), adjusted with mobile phases (A = acetonitrile and B = water:formic acid, 99.75:0.25) constant flow at 0.3 mL min^−1^. The runtime for standards (gallic acid, catechin, and chlorogenic acid) diluted in water was 17 min, using the isocratic elution mode at 5:95 *v/v* (A:B). On the other hand, a linear gradient of A:B was employed for standards (caffeic acid, ellagic acid, and quercetin) diluted in methanol: 0–8 min = 8:92; 8–14 min = 15:85; 14–22 min = 25:75. The UV absorption spectra were measured for ellagic acid at 253 nm; gallic acid and catechin at 271 nm; chlorogenic and caffeic acids at 320 nm; quercetin at 372 nm.

The analytical curve for gallic acid was built in the concentration range of 21.2–212 µg·g^−1^ (y = 9365.6x + 3731.4, R^2^ = 1). For catechin, solutions of 24.4–244 µg·g^−1^ (y = 1560.8x − 73.57, R^2^ = 0.9998) were used. The concentration range for chlorogenic acid was 18.4–184 µg·g^−1^ (y = 7684.6x + 7508.9, R^2^ = 0.9999). The ellagic and caffeic acids curves were produced at concentrations of 20–200 µg·g^−1^ (y = 22,204x − 63,030, R^2^ = 0.9999) and 20–200 µg·g^−1^ (y = 15,824x + 29175, R^2^ = 0.9999), respectively. Finally, concentrations of 21.2–212 µg·g^−1^ (y = 10,884x + 19971, R^2^ = 0.9999) were used for quercetin. The results are expressed as µg·g^−1^ extract.

### 2.13. Chemical Profile

Following the method described by Silva et al. [25], an aliquot of 2.0 µL of filtered extract (0.22 µm nylon syringe filter) was added to the edge of the chromatography paper along with methanol, acting as an ionization source, to the LCQ Fleet mass spectrometer (Thermo Scientific, San Jose, CA, USA). Positive and negative ionization modes were used in the evaluations conducted in triplicate.

The PS-MS voltage sources were +5.0 kV (positive ionization mode) and −3.5 kV (negative ionization mode); capillary voltage 40 V; transfer tube temperature 275 °C; tube lens voltage 115 V; and mass range from 100 to 1000 *m/z*. The ions with their respective fragments were identified using the literature comparisons. Furthermore, collision energies from 15 to 40 eV were used to fragment the compounds. Mass spectra obtained from this analysis were processed with Xcalibur software (Thermo Scientific, San Jose, CA, USA).

## 3. Results and Discussion

### 3.1. Characterization of Passion Fruit Peel Flour

#### 3.1.1. Color

Colorimetric parameters (L* 88.04 ± 1.25; C* 23.04 ± 1.54; h* 85.69 ± 0.67) demonstrated the clear color of the PFPF obtained in this study. Comparing colorimetric parameters obtained in the study with the literature [14,26], it was verified that our PFPF presented a lighter color. Several factors can directly interfere with the colorimetric parameters of PFPF, namely, drying time and temperature, fruit maturation, type of greenhouse, and more [27].

It was observed that the drying temperature used here (60 °C) inhibited the polyphenol oxidase enzymatic activity. This avoids the oxidation of phenolic compounds to o-quinones and, consequently, the enzymatic browning of the product [28]. Hence, these results corroborated the high TPC content in the PFPF. Considering the product’s lighter color, one may infer that our PFPF can be used in several products and food preparations without much interference in the colorimetric parameters.

#### 3.1.2. Proximate Composition and Starch Content

The water content of PFPF (9.57 ± 2.66%) was found to be in accordance with the compositional standards for flours (maximum 15% for flours derived from fruits) established by the Brazilian Health Regulatory Agency (ANVISA) [29]. The water content was similar to that of Cazarin et al. [16], which reported content of 9.48% at 50 °C. While Alves et al. [13] achieved 6.6% of water content for PFPF at 60 °C, Garcia et al. [30] obtained 6.86% when drying at temperatures between 60–105 °C.

Table 1 shows the proximal composition of PFPF as mean values ± standard deviation.

The lipid content found in this study was higher than in the literature. For instance, Hernández-Santos et al. [31] and Silva et al. [32] reported values of 0.64% and 0.63% of lipids in the PFPF, respectively. Cazarin et al. [16] and da Alves et al. [13] found even lower values for PFPF lipid content, 0.31% and 0.54%, respectively. However, it is essential to mention that using specific parts of the passion fruit residue and the method used to obtain the flour can also affect compositional aspects such as lipid content.

Regarding the protein content, the PFPF evaluated here presented 2.13 ± 0.16% protein. Although this value is similar to that found by Cazarin et al. [30] (3.94%), it is lower than those reported in other publications such as Hernández-Santos et al. [31] and Alves et al. [13], which registered protein contents of 4.62% and 7.53%, respectively. In terms of ash content, the values obtained here (Table 1) are pretty close to those reported by Oliveira et al. [28] (7.28%), Alves et al. [13] (8.07%), and Hernández-Santos et al. [31] (6.44%). Thus, it was observed that the obtaining method used in this study provided PFPF with satisfactory levels of dietary minerals. The total carbohydrate content obtained was (89.33 ± 2.66%) which was also equivalent to those found in the literature [16].

Concerning the starch content, our PFPF presented values of 3.15 ± 0.02%, close to that found in the publication by Bussolo-Souza et al. [33] (2.3%), which evaluated PFPF samples for commercial use. The similarity of starch content between the product obtained here and that reported in the literature evidences the absence of starch contamination during its production. It is essential to mention that a fraction of the starch present in the PFPF represents constituent carbohydrates of passion fruit peels such as pectin and insoluble fibers. Yet, edaphoclimatic conditions, cultivars, maturation stage, and vegetal genetics might contribute to divergences in proximal composition of PFPF. Moreover, the chemical properties of PFPF may also be influenced by the processing methods. Hence, such factors might explain varying constitutional properties found in this present work.

#### 3.1.3. Fourier Transform Infrared (FTIR) Spectroscopy

Figure 1 shows FTIR results used to determine functional groups of PFPF. As one may notice, most wave numbers were found between 1750 cm^−1^ to 900 cm^−1^ bands. The same results were observed by Canteri et al. [34], who used infrared spectroscopy to determine the composition of various fruits and vegetables. The low-intensity broad peak at the 3230 cm^−1^ band corresponds to the elongation of –OH groups found in D-glucose units [15,35]. The peak found at the 1075 cm^−1^ band, associated with v(C-O), v(C-C) of xyloglucan and phosphate, is positively correlated with phenolic compounds and non-cellulosic glucose [34,36].

The absorption peak of 1600 cm^−1^ corresponded to the C=O and -OH groups, indicating the functional groups of aldehydes, ketones, esters, and carboxylic acids, which are commonly found in fruits [37,38]. The bands at 1635 cm^−1^ and 1446 cm^−1^ are typically attributed to O=C-O functional group vibrations [39]. Bezerra et al. [40] also found these patterns for FTIR evaluations of yellow passion fruit albedo flour.

The 1735 cm^−1^ peak is related to the elongation of C=O conjugated carbonyl groups (carboxylic acid and lactone groups), corresponding to aromatic or benzene rings in the lignin [41]. The peak at 1234 cm^−1^ indicated the stretching of the lignin C-O bond [35], while the band at 1020 cm^−1^ corresponded to the C-O-C vibration of the pyranose ring [42].

Variations of spectra and lack of some bands can be observed when comparing the results presented here with the literature. These results demonstrate the variability of fruit and vegetable cell walls and the influence of material processing methods [34]. Despite this, the results found by FTIR analyses demonstrate that PFPF has lignin- and cellulose-rich composition, as expected for diverse agricultural by-products [43].

#### 3.1.4. Scanning Electron Microscopy (SEM)

The SEM technique was applied to assess the morphological surface of the developed PFPF, as presented in Figure 2.

As shown in Figure 2, PFPF presented a rough and slightly wrinkled surface (represented by the arrows in Figure 2B,C). This result may be related to thermo-sensitive compounds’ disintegration, resulting in irregular PFPF surfaces [44]. Oliveira et al. [28] found similar results for PFPF granules obtained by drying the peel at 60 °C. The irregular PFPF surface increased the surface contact, contributing to easier product dispersion in solutions and increasing the water retention in the flour [45,46]. Similar microstructure images of vegetal samples with high fiber content can be found in the literature [47]. Interestingly, the flour’s even particulate structure, with both large and small particles (Figure 2D), may contribute to its compaction, i.e., smaller particles tend to fill in the spaces left by larger ones, increasing the material’s density.

#### 3.1.5. Granulometry

As expected with flours obtained from fruit by-products, the PFPF showed a heterogeneous granulometric distribution. The 48 mesh (300 µm) and 60 mesh (250 µm), at 40.25%, and 21.29% were the highest retentions, respectively. The heterogeneous particle size distribution was also verified in the literature for PFPF obtained by several methods [48,49]. The images captured using SEM demonstrate the granulometric heterogeneity of PFPF, where it was possible to verify the flour’s particulate structure. As one may wonder, the particle size directly affects the product’s ability to absorb water, mixing time, and sensory characteristics [50].

Nevertheless, the small PFPF average particle size provided a satisfactory contact surface to enhance the release of phenolic compounds. The rupture of plant cells is characteristic of smaller particle sizes, while larger particles are particular to flours presenting high fiber contents [51].

#### 3.1.6. Total Phenolic Compounds (TPC) and Antioxidant Capacity

On average, the total phenolic compounds (TPC) content was 645.54 ± 2.47 mg EAG·100 g^−1^ of sample on a DMB. Comparing these results with other studies in the literature, the obtained PFPF stands out because of its higher content of these bioactive compounds. Morais et al. [52] evaluated the TPC content of passion fruit peel (dried and lyophilized) using methanol extraction. The authors found low levels of phenolic compounds, reporting TPC contents of 86.74–269.75 mg EAG·100 g^−1^. The low TPC contents found by the authors indicated that heating the samples helped break down plant membranes and increased the phenolic compounds’ bioaccessibility [26]. Similarly, Nascimento et al. [53] detected TPC contents in the range of 143.8–381.3 mg EAG. 100 g^−1^ in extracts from yellow passion fruit peels, in both natural and dry samples.

The divergence of TPC contents is associated with several factors, such as methods and process conditions, which may affect the levels of bioactive compounds. Therefore, the same material may present different results regarding the quantification of TPC [53,54], depending on the methods used to process the raw material.

Plants are natural sources of many polyphenolic compounds with different actions and polarities. Hence, using more than one method for evaluating their antioxidant capacities is essential. Thus, various mechanisms of action can be assessed [16,27]. In this study, the PFPF antioxidant capacity was carried out using three methods, where 29.06 ± 1.70 μM of Trolox·g^−1^ of sample on a DMB was measured using ABTS, 72.38 ± 2, 66 μM of ferrous sulfate·g^−1^ of sample on a DMB with FRAP, and 79.05 ± 1.65 μM of Trolox·g^−1^ of sample on a DMB was registered using DPPH. Cazarin et al. [16] determined the antioxidant capacity of aqueous, methanolic/acetonic, and ethanolic extracts of passion fruit peels dehydrated at 50 °C. The reported detected antioxidant capacities ranged from 29.6 ± 0.66 to 46.35 ± 0.85% (DPPH method) and 34.95 ± 2.02–38.65 ± 1.55 μmol Trolox·g^−1^ (FRAP method). When evaluating fiber concentrates obtained from dry passion fruit residues, Martínez et al. [55] determined the antioxidant capacity of radical scavenging ABTS equal to 2.4 ± 0.25 and 5.5 ± 0.58 μM of Trolox·g^−1^ for passion fruit residue extracts obtained with ethanol and methanol–acetone, respectively.

It can be inferred that the antioxidant capacity is also influenced by the cultivar, maturation stage, flour production method, as well as the compounds extraction conditions [56]. In addition to bioactive compounds, studies have shown that PFPF pectin and fibers can effectively sequestrate free radicals such as DPPH and ABTS [57,58].

The PFPF extract eliminated the DPPH radical more efficiently, indicating the excellent hydrogen donation capacity and evidencing the strong antioxidant capacity of this residue [59]. Therefore, given the results found in this study, PFPF can be used to obtain foods and products with antioxidant properties.

#### 3.1.7. Carotenoids

The PFPF carotenoid content of 645.06 μg carotenoid·100 g^−1^ on a DMB, was significantly higher than other vegetable by-products. The publication by Chutia and Mahanta [60] reported slightly higher results of carotenoid content evaluated for PFPF obtained from freeze-dried passion fruit peel (739.456 µg·100 g^−1^). These results pointed out the vital role temperature might play during the carotenoid extraction process since samples subjected to high drying temperatures, such as those obtained in this study, presented more elevated amounts of carotenoids [27,60,61]. Therefore, higher temperatures made carotenoids more available [26].

Carotenoids are precursors of vitamin A. This valuable nutrient offers antioxidant properties and favors strengthening the immune system, promoting numerous health benefits [62]. Therefore, because of its considerable content of carotenoids, PFPF may be used in developing and elaborating functional food products.

#### 3.1.8. Determination of the Profile of Phenolic Compounds

Table 2 shows the groups where phenolic compounds found in the evaluated PFPF are classified.

Figure 3 shows the chromatogram of the phenolic compounds identified in the extract obtained from the PFPF.

As one may notice from Table 2, phenolic acids (gallic acid, caffeic acid, and ellagic acid) and flavonoids (quercetin) were the major groups identified. The phenolic acids identified in PFPF can be ordered in descending concentrations: gallic acid > caffeic acid > ellagic acid > quercetin. It is essential to mention that chlorogenic acid and catechin were not detected in this analysis.

The phenolic compounds found in this study were also identified in other publications regarding the PFPF composition assessment [16,26,63,64]. However, the lower amount of quercetin determined here may be explained by its instability at higher temperatures, which might decrease its concentration in the evaluated samples [26]. Furthermore, the non-detection of chlorogenic acid and catechin can be justified by different conditions such as the location of fruit’s cultivation, variety, maturation stage, harvesting conditions, and the extraction method. The influence of the type of solvent used can also be mentioned since the solvent’s polarity interaction with the sample may directly affect the extraction yield [54].

#### 3.1.9. Chemical Profile

The fingerprints of the chemical compounds of the PFPF extract are shown in the Figure 4.

The identification of compounds using PS-MS negative ionization mode distinguished organic, phenolic acids, fatty acids, phenylpropane glycerides, sugar, flavonoids, and quinone, summing up 13 substances. Table 3 shows the chemicals identified.

As one may notice, most of the PFPF chemical compounds were classified within the phenolic acid group. Considering this rich bioactive compound composition, one may infer that this by-product has enormous potential to be evaluated in terms of its in vivo performance, as evidenced by Reis et al. [57], who studied the antioxidant potential of *P. edulis*. Differently from chlorogenic acid, caffeic acid was also detected by UPLC.

Regarding the results obtained by MS/MS analyses, it was proposed that the ion identified by *m/z* 133 can be classified as malic acid [M-H]-, based on its MS/MS transition 133 → 115 (loss of a water molecule), which was also observed by Silva et al. [56], who studied the chemical composition of banana peel flour samples. It is essential to mention that this present study was a pioneer in identifying malic acid in PFPF. Concerning the phenylpropane glycerides chemical class, the compound 1-*O*-dihydrocafeoyl glycerol (*m/z* 255) identified here was also identified in sorghum grains and banana peel flour [56,66]. The signal at *m/z* 215, identified as a hexose, comprises the PFPF high carbohydrate content, in accordance with results presented in Table 1.

Concerning the flavonoid group, six compounds were identified using both ionization modes. Remarkably, flavonoids are suggested to be compounds of passion fruit with therapeutic properties [58,71]. Oleic and stearic acids, some of the main saturated and unsaturated fatty acids in the passion fruit seed [58], were also detected in the peel. Oleic acid was also the most abundant fatty acid found in the dry aerial parts of *P. edulis* (leaf and stem) by Otify et al. [71].

Publications such as that by Ramli et al. [72] also applied PS-MS to evaluate passion fruit peel extract’s phenolic and flavonoid contents, detecting 11 compounds; this is fewer components than the number found in this research. These identified variations (types and amount of compounds) may be related to several factors, ranging from the fruit’s variety, degree of maturation, cultivation conditions, and extraction methods used.

Table 4 presents PFPF compounds identified using the positive ionization mode, summing up nine different molecules and comprising diverse chemical classes, namely, flavonoids, sugars, fatty acids, terpenes, and amino acids.

The primary compound group identified was flavonoids, usually conjugated as a form of protection in plants and the largest group of natural phenolics present in vegetables [56,73]. As in the negative mode, it was possible to detect different sugar molecules. Vomifoliol β-D-glucopyranoside was identified as a major ion at *m/z* 409 [M + Na]+ and a fragment derived at *m/z* 247, which is attributable to the loss of a glucosyl residue ([M + Na −162]+). The ion from *m/z* 391 was formed by the successive loss of H2O from *m/z* 409, which was also observed by Jia et al. [74]. Correia et al. [73] evaluated sorghum flour and also identified the presence of the fatty acid cnidioside methyl ester at *m/z* 413, which is a fatty acid ester that can be produced by an alkali-catalyzed reaction between fatty acids and methanol.

## 4. Conclusions

The passion fruit peel flour (PFPF) obtained in this study presented high contents of total phenolic compounds (645.54 mg EAG·100 g^−1^ on a DMB), carotenoids (645.06 μg carotenoid·100 g^−1^ on a DMB), and interesting antioxidant capacities, measured by complementary technique, namely, ABTS (29.06 μM of Trolox·g^−1^ on a DMB), FRAP (72.38 μM of ferric sulfate·g^−1^ on a DMB), and DPPH (79.05 of Trolox·g^−1^ on a DMB). The Ultra-Performance Liquid Chromatography (UPLC) analyses were used to identify and quantify four different components in the PFPF samples, which were present in descending concentrations as follows: gallic acid > caffeic acid > ellagic acid > quercetin. The paper spray mass spectrometry (PS-MS) identified nine chemical compounds using the positive ionization mode, covering the following compound classes: flavonoids, sugars, fatty acids, terpenes, and amino acids. On the other hand, PS-MS negative mode identified 13 compounds from different chemical groups: organic, phenolic, and fatty acids, phenylpropane glycerides, sugars, flavonoids, and quinone. The PFPF samples evaluated here presented granulometric heterogeneity in accordance with Scanning Electron Microscopy (SEM) analysis, which demonstrated the samples’ rough and particulate surface. The results obtained by Fourier transform infrared (FTIR) showed that lignin, alcohols, aldehydes, ketones, carboxylic acids, phenolic compounds, and esters are the functional groups found in PFPF samples.

Considering this study’s findings, we infer that the obtained PFPF has interesting constitutive properties besides its rich phenolic compound composition and high antioxidant capacity. Therefore, the flour obtained from the passion fruit peel, usually regarded as an agro-industrial waste, offers potential for applications as a natural ingredient in various food product formulations.

## Figures and Tables

**Figure 1 metabolites-13-00684-f001:**
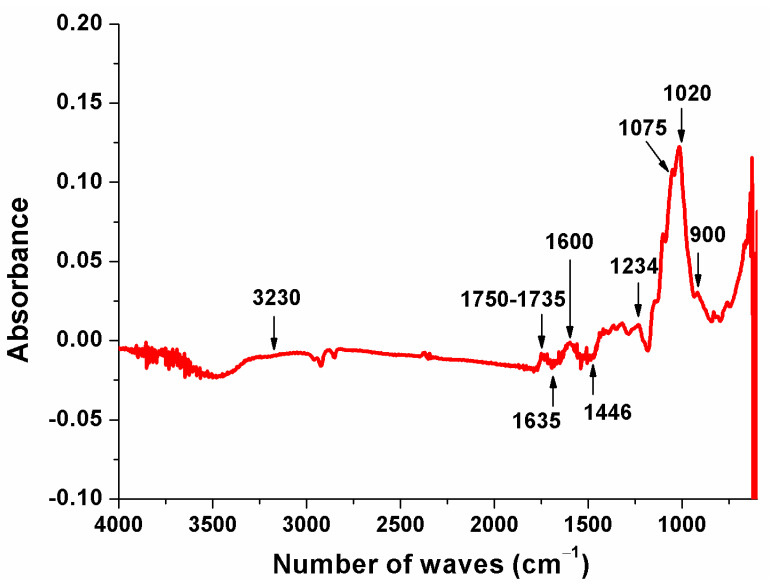
Fourier transform infrared spectrum of the PFPF.

**Figure 2 metabolites-13-00684-f002:**
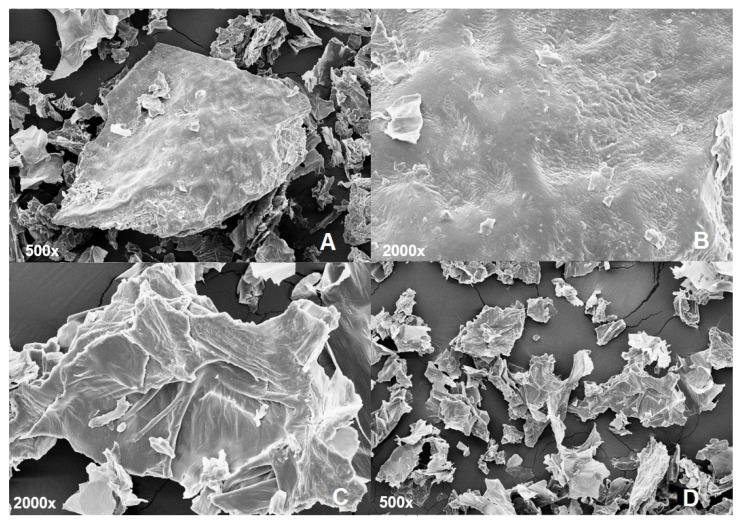
Scanning electron microscopy images of flour obtained from passion fruit peel.

**Figure 3 metabolites-13-00684-f003:**
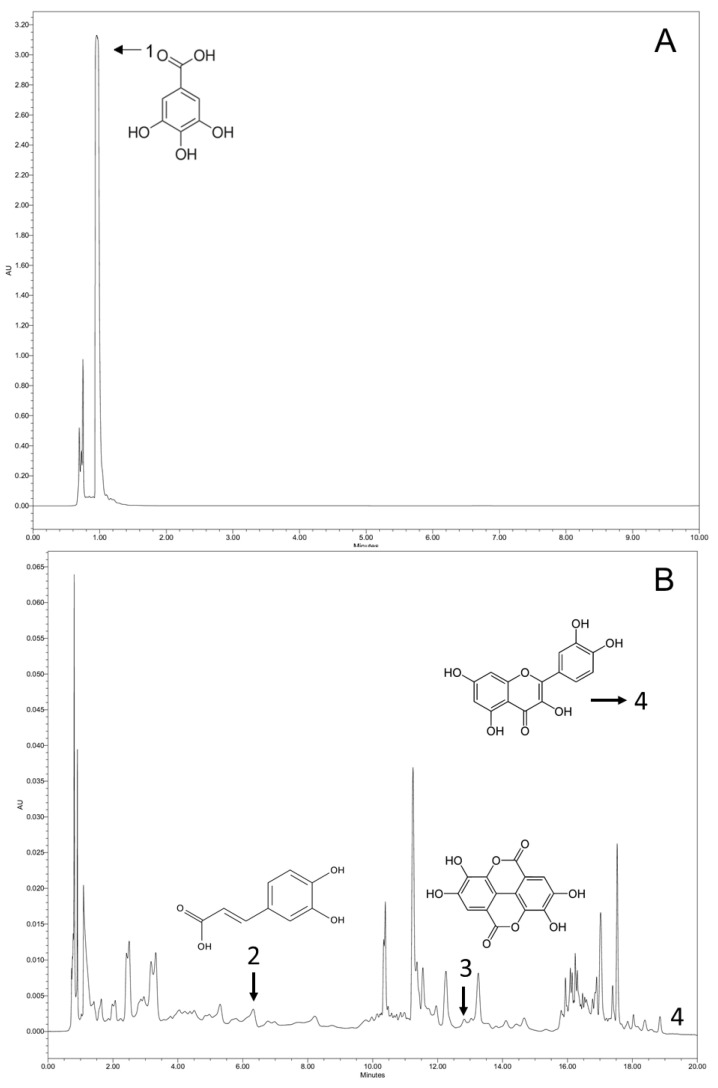
Chromatogram of phenolics identified in the extract obtained from PFPF at 320 nm. (**A**) Spectrum of standards diluted in water; 1: gallic acid; (**B**) spectrum of standards diluted in methanol; 2: caffeic acid; 3: ellagic acid; 4: quercetin.

**Figure 4 metabolites-13-00684-f004:**
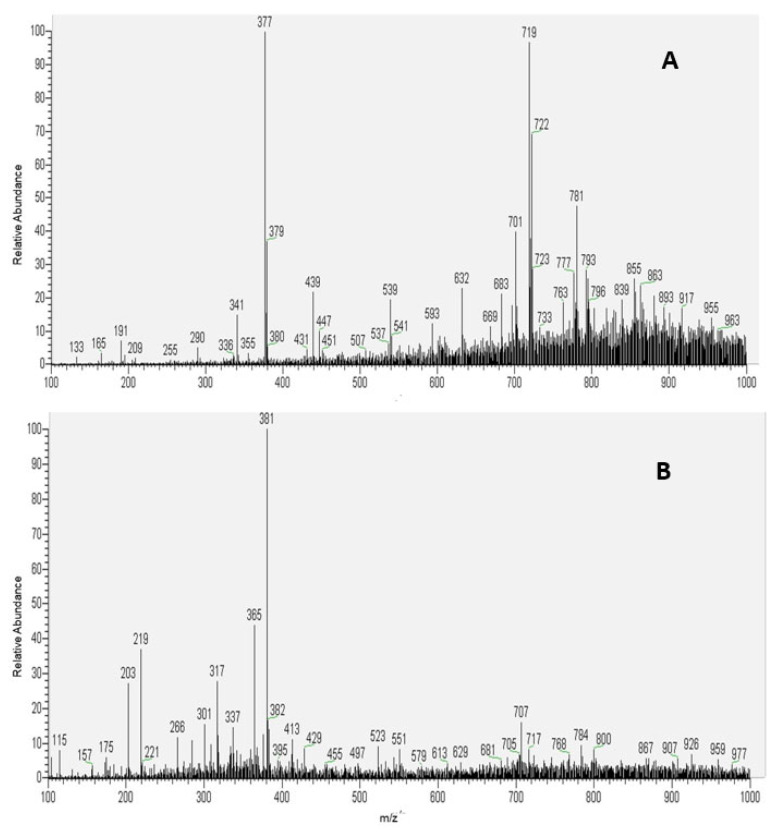
Representation of mass spectra by paper spray mass spectrometry (PS-MS). (**A**) PS-MS spectrogram in negative ionization mode and (**B**) PS-MS spectrogram in positive ionization mode of extract obtained from passion fruit peel flour.

**Table 1 metabolites-13-00684-t001:** Mean values of the proximal composition of PFPF and starch content, on a dry matter basis.

Components	Content * (%)
Lipid	1.24 ± 0.20
Protein	2.13 ± 0.16
Ash	7.28 ± 0.11
Total carbohydrates	89.33 ± 2.66
Starch	3.15 ± 0.02

* Mean value ± standard deviation.

**Table 2 metabolites-13-00684-t002:** Profile of phenolics (μg·g^−1^ of extract) in the extract for PFPF.

Compound	PFPF Content *
Gallic acid	12.37 ± 0.01
Caffeic acid	2.32 ± 0.02
Ellagic acid	0.61 ± 0.02
Quercetin	0.34 ± 0.01
Chlorogenic acid	ND
Catechin	ND

* Mean value ± standard deviation. ND: not detected.

**Table 3 metabolites-13-00684-t003:** Ions identified in passion fruit peel flour by PS-MS using the negative mode.

Identification	*m/z*	MS/MS	Reference
Organic acid			
Malic acid	133	115	[56,65]
Phenolics acid			
Caffeic acid	179	135	[66]
Caftaric acid	311	133	[65]
Chlorogenic acid	353	191	[67]
3,5-Di-*O*-caffeoylquinic acid methyl ester	535	445	[56]
Phenylpropanoid Glycerides			
1-*O*-dihydrocaffeoyl glycerol	255	163	[56,66]
Sugar			
Hexose	215	89, 179	[56,68]
Flavonoids			
Luteolin-7-glucoside (flavone)	447	227	[69]
Kaempferol-3-*O*-rutinoside	593	285, 447	[56,70]
Rutin (quercetin 3-rutinoside)	609	341	[56,70]
Fatty acids			
Oleic acid	281	237	[69]
Stearic acid	283	265	[69]
Quinone			
4,9-Dihydroxy-6,7-dimethoxynaphtho(2,3-d)-1,3-dioxole-5,8-dione	293	249	[69]

**Table 4 metabolites-13-00684-t004:** Ions identified in passion fruit peel flour by PS-MS in positive mode.

Tentative Identification	*m/z*	MS/MS	Reference
Flavonoids			
Diosmetin	301	258	[71]
3-O-methylquercetin	317	245, 273, 247	[73]
quercetin-3-malonylglucoside	551	303	[66]
Sugars			
Sucrose	381	219	[21]
Morroniside	429	267	[74]
Fatty acid			
Cnidioside Methyl Ester	413	413	[74]
Terpene			
Deacetylforskolin	369	253	[64]
Vomifoliol β-D-glucopyranoside	409	394, 391, 247	[74]
Amino acids			
L-arginin	175	129	[21]

## Data Availability

Not applicable.

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
