# Peer review of "Development and Characterization of Yellow Passion Fruit Peel Flour (Passiflora edulis f. flavicarpa)"

_metabolites, 2023, doi:10.3390/metabo13060684_

Round 1

Reviewer 1 Report

The aim of this research study was to evaluate the flour obtained from the peels of the yellow passion fruit. The physicochemical, microscopic, colorimetric, granulometric characteristics, content of phenolic compounds, carotenoids, antioxidant activity, presence of functional groups by infrared spectroscopy, profile of phenolic compounds by liquid chromatography and chemical profile through paper spray mass spectrometryof the flour produced were evaluated.

This research study is interesting because currently several studies have been associated plant-derived products with various physiological benefits, including antioxidative, immunomodulatory, anti-tumor, anti-inflammation, antidiabetic, and antiviral properties. Furthermore, the production of this flour from passion fruit peel would avoid eliminating these by-products and take better advantage of its nutritional and antioxidant components, thus reducing waste of resources and environmental problems. In general, manuscript is clearly describing the scientific findings with related experimental data. The significance of the study is sound, with all the data being clearly presented.

Major comments

1) Please provide in the introduction section a description of the botanical characteristics (genus and family minimun),of the passion fruit.

2) In the materials and methods section, I would like the authors to describe the reagents and laboratory chemicals used in this study, as well as the company where they were purchased.

In the sentence lines 63-64 “Passion fruit peels Passiflora edulis f. Flavicarpa were acquired in a popular market the city of Sete Lagoas-MG, in June 2019.” MG refers to Minas Gerais?

3) Has any taxonomic classification been carried out to determine that it is indeed passion fruit peel?

4) In-text citations should be according to the journal metabolites, please check them. For example on page 2 lines 88 and 89. In-text citations should be according to the journal, please check them.

For example on page 2 lines 88 and 89. The sentence " The starch content was determined according to [15]" should be "The starch content was determined according to BRASIL [15]".

Consider using another citation as citation number 15 cannot be verified. Please check them carefully  

-Please unify the abbreviations. For example In the lines 92-93, the authors describe “with a DLaTGS detector (deuterated triglycine sulfate doped with L-alanine)” in my opinion it should be “with a deuterated triglycine sulfate doped with L-alanine (DLaTGS) detector”

-Please could you tell me what "ABNT" means?

- Please describe further in section "2.9. Total phenolic compounds and antioxidant activity". I attach an example from which you can be guided:

Mateos et al. (2022). Identification, Quantification, and Characterization of the Phenolic Fraction of Brunfelsia grandiflora: In Vitro Antioxidant Capacity. Molecules 27(19), 6510. https://doi.org/10.3390/molecules27196510

-Please describe in more detail section "2.11. Determination of the profile of phenolic compounds".

-Again, please uniform the references in the text. Line 151 “SILVA et al. [21]” or “Silva et al. [21]”

5) In the section “3. Results and Discussion”. Please be very careful, I do not see Table 1 in the text.

- Please be very careful with the numbering of tables in the text.

- I think it is not necessary to mention the “Statistical Analysis” section as throughout the results and discussion there is no statistical analysis per se. Rather, it compares the results with other studies.

It is important to mention that the Discussion should be improved with current references and considering journals with a relative quality index.

Minor comments

- I would also suggest that the author revised the grammar of the manuscript to increase the English level of the manuscript.

- Please adapt the references according to the journal metabolites.

- Please uniform the units according to the journal metabolites

- I suggest you update the references, mostly cited to local journals. So I recommend that you update them with other repositories.

Recommendation

For the reasons stated above, I recommend that this work be Reconsider after major revision  and  I hope the outcome of this specific submission will not discourage you from the submission of future manuscripts.

Author Response

Reviewer #1:

1) Please provide in the introduction section a description of the botanical characteristics (genus and family minimun), of the passion fruit.

Response: Lines 50 - 51 – A description of the botanical characteristics of the passion fruit genus and family was added.

2) In the materials and methods section, I would like the authors to describe the reagents and laboratory chemicals used in this study, as well as the company where they were purchased.

Response: Lines 83 - 93 – The reagents and chemicals used in this study, as well as the company from which they were purchased, have been added.

3) In the sentence lines 63-64 “Passion fruit peels Passiflora edulis Flavicarpa were acquired in a popular market the city of Sete Lagoas-MG, in June 2019.” MG refers to Minas Gerais?

Response: Line 96 – Yes, it refers to the state of Minas Gerais. Added full state name in sentence.

4) Has any taxonomic classification been carried out to determine that it is indeed passion fruit peel?

Response:  The taxonomic classification was not carried out, however, the fruit used to make the flour is actually the passion fruit. This fruit is widely cultivated and consumed in Brazil, easily purchased in the markets. Thus, the peel was obtained directly from the fruits purchased in the local market.

5) In-text citations should be according to the journal metabolites, please check them. For example on page 2 lines 88 and 89. In-text citations should be according to the journal, please check them. For example on page 2 lines 88 and 89. The sentence " The starch content was determined according to [15]" should be "The starch content was determined according to BRASIL [15]". Consider using another citation as citation number 15 cannot be verified. Please check them carefully.

Response: All citations throughout the manuscript have been reviewed and appropriated according to Metabolites Journal guidelines. Line 121 - Citation 15 was changed to a citation using an international method.

6) Please unify the abbreviations. For example In the lines 92-93, the authors describe “with a DLaTGS detector (deuterated triglycine sulfate doped with L-alanine)” in my opinion it should be “with a deuterated triglycine sulfate doped with L-alanine (DLaTGS) detector”

 Response: All abbreviations were unified and  the DLaTGS nomenclature (Lines 125 – 126) was changed.

7) Please could you tell me what "ABNT" means?

Response: Line 141 – ABNT means "Brazilian Association of Technical Standards". This term was inserted in the manuscript.

8) Please describe further in section "2.9. Total phenolic compounds and antioxidant activity". I attach an example from which you can be guided.

Response: Line 153 – 191. Detailed description was performed.

9) Please describe in more detail section "2.11. Determination of the profile of phenolic compounds".

Response: Line 204 – 225. Detailed description was performed.

10) Again, please uniform the references in the text. Line 151 “SILVA et al. [21]” or “Silva et al. [21]”

Response: Line 227 – This reference and all others were revised and standardized throughout the entire manuscript.

11) In the section “3. Results and Discussion”. Please be very careful, I do not see Table 1 in the text. Please be very careful with the numbering of tables in the text.

Response: Line 263. Table 1 was inserted in the text and all the numbering of all tables and figures in the text was verified.

12) I think it is not necessary to mention the “Statistical Analysis” section as throughout the results and discussion there is no statistical analysis per se. Rather, it compares the results with other studies.

Response: Lines 239 – 487. Statistical mention was removed and the results obtained in this work were compared with other studies.

13) It is important to mention that the Discussion should be improved with current references and considering journals with a relative quality index.

Response: Lines 239 – 487.  The discussion was improved as much as possible, as most of the articles found in the literature on passion fruit flour were published in journals with low quality scores.

Minor comments

- I would also suggest that the author revised the grammar of the manuscript to increase the English level of the manuscript.

Response: Manuscript grammar was revised throughout the manuscript.

- Please adapt the references according to the journal metabolites.

Response: All references were reviewed and adapted according to the journal's guidelines.

- Please uniform the units according to the journal metabolites

Response: All units have been revised and standardized according to Metabolites Journal standards

- I suggest you update the references, mostly cited to local journals. So I recommend that you update them with other repositories.

Response: Updated references with a higher impact factor were added as far as possible, since most works on passion fruit were published in lower impact journals.

Recommendation

For the reasons stated above, I recommend that this work be Reconsider after major revision  and  I hope the outcome of this specific submission will not discourage you from the submission of future manuscripts.

All suggestions were accepted and the manuscript was thoroughly revised. Thank you once again for your valuable comments. We appreciate the time and effort spent in this reviewing process.

Reviewer 2 Report

The intruduction should be revised, more related works about the passion fruit residue flour or similar plant should be exhibited. The differences of your work and references should be described. The structures of bioactive compounds from passion fruit residue flour should be added. 

Author Response

Reviewer #2:
1) The introduction should be revised, more related works about the passion fruit residue flour or similar plant should be exhibited. The differences of your work and references should be described.

Response: Lines 39 – 80. The entire introduction was revised and more works related to passion fruit flour and the comparison with the work presented here were added.

2) The structures of bioactive compounds from passion fruit residue flour should be added. 

Response: Thanks for the sugestion. Supplementary material with the structure of compounds has been added.

All suggestions were accepted and a thorough review was carried out throughout the manuscript. Thank you once again for your valuable comments. We appreciate the time and effort spent in this reviewing process.

Reviewer 3 Report

The title of Manuscript should be reframed and reduce the words.

Line 16-20: Split 1 sentence in 2-3 lines

Line 43: Add how much of waste generated from peel in Brazil annually with recent reference.

Line 44: 703,489 tons in 2019, add atleast 2021 data if possible add 2022 data of production of P. edulis in Brazil and also add worldwide production annually in recent years

All the figure having very low resolution, improve all figures resolution

Whole MS is in plagiarism i.e Abstract, Material method, Result discussion and conclusion

Reduce the % of plagiarism upto 10%

All genus and species name should be be in Italics

The writing could be improved by strengthening the connectivity between paragraphs. There are several places where new topics are introduced and connections to the previous subject are not clear. Read whole manuscript and correct wherever required.

Introduction:

The introduction does not clearly state the purpose of the research – please amend.

Conclusions

The conclusions are too general, format according to future aspects. Please make them more specific.

Carefully read whole manuscript line by line and improve the sentence formation

Cross check all references and style of reference according to Journal format, use abbreviation of journal name in reference

Author Response

Reviewer #3:

1) The title of Manuscript should be reframed and reduce the words.

Response: Lines 2 - 3 – The title of the manuscript was reformulated and the words reduced

2) Line 16-20: Split 1 sentence in 2-3 lines.

Response: Line 19 – 22. The sentence was divided into 2 lines.

3) Line 43: Add how much of waste generated from peel in Brazil annually with recent reference.

Response: Lines 54 – 59.  No data were found regarding the amount of passion fruit peel residue generated in Brazil, but an estimate was made based on the proportion of the peel weight in relation to its total weight, and the % destined for the production of juices and pulps, which is the process responsible for the disposal of peels.

4) Line 44: 703,489 tons in 2019, add atleast 2021 data if possible add 2022 data of production of edulisin Brazil and also add worldwide production annually in recent years

Response: Lines 54 – 55.  Added the most recent data (2021) on passion fruit production in Brazil, released by the Brazilian Institute of Geography and Statistics (IBGE). World production data were not found annually in recent years, only in 2017 released by FAO - Food and Agriculture Organization of the United Nations estimated in 1.5 million tons in 2017 released in 2018. Until now FAO has not released data on world production of passion fruit in the years following 2017.

5) All the figure having very low resolution, improve all figures resolution

Response: The resolutions of all figures have been improved.

6) Whole MS is in plagiarism i.e Abstract, Material method, Result discussion and conclusion. Reduce the % of plagiarism upto 10%

Response: Thanks for the guidance on this issue.

7) All genus and species name should be be in Italics

Response: All species names and genera have been checked and written in italics.

8) The writing could be improved by strengthening the connectivity between paragraphs. There are several places where new topics are introduced and connections to the previous subject are not clear. Read whole manuscript and correct wherever required.

Response: The entire manuscript was read again and we tried to correct connectivity between paragraphs as well as between sentences as well.

9) Introduction: The introduction does not clearly state the purpose of the research – please amend.

Response: Lines 39 – 79. The introduction was reformulated and the objectives of the manuscript were rewritten in order to meet the suggestion.

10) The conclusions are too general, format according to future aspects. Please make them more specific.

Response: Lines 487 – 509. The entire conclusion has been rewritten to meet the suggestion.

11) Carefully read whole manuscript line by line and improve the sentence formation

Response: An attempt was made to improve sentence formation throughout the manuscript.

12) Cross check all references and style of reference according to Journal format, use abbreviation of journal name in reference

Response: All references and style references have been checked to meet Metabolites Journal standards.

All suggestions were accepted and a thorough review was carried out throughout the manuscript. Thank you once again for your valuable comments. We appreciate the time and effort spent in this reviewing process.

Round 2

Reviewer 1 Report

The authors have significantly improved this research article and have followed the indications/suggestions of the other reviewers. However, I should like to make some minor observations that need to be corrected.

- Line 82, Page 2: Please enter correct chemical formula "FeCl3·6H2O" with subscripts 

- Line 112, Page 3: Please place the abbreviation "AOAC" in brackets.

- Lines 118-119, Page 3: Please consider change the following sentence "The results were expressed in g/100 g sample by dry basis (d.b.). to "The results were expressed in g/100 g sample by dry matter basis (DMB). " I think "dry matter basis (DMB)" is a more appropriate phrase than "dry basis (d.b.)"

- Lines 160-161, Page 4: The following sentence "The results were expressed as gallic acid equivalents (mg GAE.g-1 of sample by dry basis (d.b.))." If they decide to change to "dry matter basis (DMB)", only the abbreviation should be used as follows "The results were expressed as gallic acid equivalents (mg GAE.g-1 of sample by DMB)." 

- Lines 168-169, Page 4: Same problem as above, please consider using the following "...trolox equivalents (μM TE.g-1 of sample per DMB)".

- Lines 178, 179, 188 and 189, Page 4: Please correct according to my previous suggestions

-In sentences in the text where "dry basis (d.b.)" exists, it should be replaced by "dry matter basis (DMB)"

-Please revise the manuscript again, for example on line 212 and 219, page 5 "ellagic acid, and quercitin" should be "gallic acid, and quercetin". Please check the text thoroughly again as well as the English grammar.

Author Response

All suggestions were accepted and a thorough review was carried out throughout the manuscript. Thank you once again for your valuable comments. We appreciate the time and effort spent in this reviewing process.

Reviewer 2 Report

The manuscript can be accepted in Metabolites

Author Response

Thank you for your comments.

Reviewer 3 Report

Accepted in present form.

Author Response

Thank you for your comments.